# Reduction of Membrane Lipid Metabolism in Postharvest Hami Melon Fruits by *n*-Butanol to Mitigate Chilling Injury and the Cloning of Phospholipase D-β Gene

**DOI:** 10.3390/foods12091904

**Published:** 2023-05-06

**Authors:** Shuai Huang, Ying Bi, Hui Li, Caihong Liu, Xue Wang, Xinyu Wang, Yaxin Lei, Qi Zhang, Jing Wang

**Affiliations:** College of Food Science and Pharmacy, Xinjiang Agricultural University, Urumqi 830052, China; h15099211440@outlook.com (S.H.); 13238150177@163.com (Y.B.); lhui911@163.com (H.L.); liucaihong1221@163.com (C.L.); wx10080303@163.com (X.W.); 13111162656@163.com (X.W.); 17885706533@163.com (Y.L.); 18799696721@163.com (Q.Z.)

**Keywords:** *n*-butanol, Hami melon, chilling injury, membrane lipid metabolism, gene cloning

## Abstract

To investigate the effect of *n*-butanol on postharvest membrane lipid metabolism of Hami melon (Cucumis melo ‘Hami’), the fruits were soaked in a 1.0% solution of *n*-butanol for 30 min with water as the control. Symptoms of chilling injury were observed regularly, and the indices related to permeability and membrane lipid metabolism of pericarp cells were measured. The results showed that treatment with *n*-butanol inhibited the increase in chilling injury index, membrane permeability, and malondialdehyde content of Hami melon fruits, promoted an increase in the contents of phosphatidyl alcohol and unsaturated fatty acids, such as linoleic acid, linolenic acid, oleic acid (except 14 d), and erucic acid (28–42 d), and decreased the content of saturated fatty acids, stearic acid (0–28 d), phosphatidic acid (except for 21 d), and the key enzymes of membrane lipid metabolism compared with the control. The activities of phospholipase D (PLD) and lipoxygenase (LOX) and the downregulation of the levels of expression *CmPLD-β* and *CmLOX* (42 d only) genes reduced the chilling injury index of Hami melon and alleviated the further expansion of chilling injury symptoms in the fruits. We also cloned the key gene of membrane lipid metabolism *CmPLD-β*, which was obtained by pre-transcriptome screening of the pericarp. We found that *CmPLD-β* of Hami melon had the closest affinity with cucumber (CsXP5), indicating that the *CmPLD-β* gene of Hami melon was functionally similar to that of cucumber. In addition, a two-fold alignment analysis of *CmPLD-β* and CmXP5 base sequences indicated that the base sequences of the two promoter regions differed from each other.

## 1. Introduction

Xinjiang Hami melon (*Cucumis melo* ‘Hami’) as a local specialty fruit is favored by the majority of fruit lovers and is now ranked as the top fruit in Xinjiang [1]. Owing to the concentrated postharvest period of Hami melon, it is a typical respiratory leap-type fruit. In production, low-temperature storage is usually used to extend its freshness period, but Hami melon is sensitive to a low-temperature environment. Therefore, unsuitable low temperatures render the fruit prone to chilling injury, and chilling injury symptoms initially appear as dark or light brown spots on the surface, followed by sunken spots on the tissue epidermis. Thus, it is highly important to study the regulatory technology of the tolerance of Hami melon to low temperatures after harvest to extend its storage and preservation.

The cell membrane is an interface structure that initially senses cold stress. Under the conditions of chilling injury, fruits and vegetables show symptoms owing to a decrease in the content of unsaturated membrane lipids, increased membrane permeability, leakage of intracellular electrolytes and soluble substances, a gradual decrease in enzyme activity bound to the membrane, and the dysregulation of enzymatic reactions that lead to cell membrane damage [2]. The lipids in cell membranes are composed of three main components, including phospholipids, glycolipids, and cholesterol. Phospholipids are the most important component and play a pivotal role in signal transduction, membrane flow, and cytoskeleton composition [3]. Phosphatidic acid (PA), phosphatidylcholine (PC), phosphatidylethanolamine (PE), phosphatidylglycerol (PG), and phosphatidylserine (PS) are present in the structure of phospholipid fractions of plant cells. These phospholipids can undergo hydrolysis and transfer under the action of phospholipase, thus changing the structure and function of cell membranes. When plants receive stress signals, phospholipids in the cell membrane are degraded, and the structure and integrity of the cell membrane are disrupted. In addition, phospholipids are not only the primary constituents of cell membranes but also participate as signaling molecules in various activities of plant metabolism, and PA can directly stimulate antioxidant enzymes and activate the expression of abscisic acid to stimulate the response of plant resistance mechanisms [4]. In addition, phospholipids are attached to a variety of unsaturated fatty acid chains, which together with phospholipids perform the functions of cell membranes. Common fatty acids in plants include myristic, palmitic, arachidonic, stearic, yaconic, oleic, lauric, linoleic, and linolenic acids among others. When plants are subjected to stress or during aging, phospholipases are stimulated to hydrolyze the unsaturated fatty acids to produce polyunsaturated fatty acids; this is followed by the oxidation of unsaturated fatty acids by lipoxygenase (LOX) and a decrease in the unsaturation index, which leads to a decrease in the fluidity of cell membrane [5].

Phospholipase D (PLD) is also a key enzyme in phospholipid metabolism, which can hydrolyze a variety of phospholipids, such as PC, PI (phosphatidylinositol), PE, and their derivatives to produce PA and inositol triphosphate (IP3) [6]. It has been shown that in plants, phospholipase D has physiological functions involved in signaling the formation of biofilms, stress resistance, and cellular lipid metabolism [7], and there are several gene families of PLD with PLDα the most common [8]. The involvement of CaPLDa4 in the degradation of membrane lipids has been shown under low-temperature stress, and it affects chilling injury in green pepper fruits (*Capsicum annuum*) [9]. The high level of expression of CmPLD in cucumber (*Cucumis sativus*) fruits activates PLD and LOX activities and accelerates chilling injury in fruits [6]. The activity of PLD increases in “banana plum” fruits after low-temperature stress, and the expression of PLDα subfamily members is upregulated. The expression of PLDα subfamily members was upregulated; PLD activity was activated, LOX activity, malondialdehyde (MDA) content, and cell membrane permeability also increased; the fraction of PA in the membrane lipids increased; the contents of PE, PC, and PI all decreased, and the cell membrane suffered damage [10]. The occurrence of chilling injury in peach (*Prunus persica*) fruit was related to PpPLD activity because all the PpPLD was activated at 28–35 d of storage [11]. A correlation between low temperature and PLD activity and the expression of its gene was also found in research on green pepper [9], loquat (*Eriobotrya japonica*) [12], and strawberry (*Fragaria x ananassa*) [13]. Identifying how to regulate PLD activity to reduce the degradation of membrane lipids and maintain the integrity of fruit cell membrane structure in response to low-temperature stress is a highly active area of current research.

It has been shown that *n*-hexanal, *n*-butanol, and *sec*-butanol, inhibitors of PLD activity, have been applied to preserve postharvest fruits, such as longan (*Dimocarpus longan*) [14], peach [11], and mulberry (*Morus alba*) [15]. However, *n*-butanol is a specific inhibitor of phospholipase D. The presence of *n*-butanol results in quicker reactions of PLD in transphosphatidylation, thus inhibiting the hydrolytic activity of PLD [16]. The treatment of peach fruit with *n*-butanol inhibits the MDA produced by the peroxidation of membrane lipids, cell membrane permeability, and the expression of *PpPLDa*, which enhanced its tolerance to cold [11]. The treatment of Nanguo pear (*Pyrus ussuriensis*) fruit with *n*-butanol increased the activities of PLD and LOX and reduced the levels of expression of *CmPLD* and *CmLOX* in the fruits [17]. A related study found that the treatment of litchi (*Litchi chinensis*) with *n*-butanol maintained a high content of ascorbic acid and reduced chilling injury, mold activity, browning, and the activity of PLD in frozen litchi [18]. To our knowledge, research on how *n*-butanol regulates membrane lipid metabolism in low-temperature frozen Hami melon fruits and mitigates chilling injury has scarcely been reported.

Therefore, in this study, we investigated the effect of treatment with *n*-butanol on the postharvest membrane lipid metabolism of Hami melon fruits and cloned the *CmPLD-β* gene from their pericarps to investigate its mechanism of response to cold stress and provide a basis for the cold storage of Hami melon fruit and cloning of the *CmPLD-β* gene.

## 2. Materials and Methods

### 2.1. Materials and Reagents

“Xizhou Mi 25” Hami melon was picked on 21 July 2019 from the commercial melon base of the 103rd regiment in Wujiaqu City, Xinjiang Uygur Autonomous Region, China. Melons of basically the same maturity and uniform size that were free from pests and diseases, decay, and mechanical damage were selected for picking, and each melon was packed individually using mesh foam immediately after picking. The fruits were then boxed in special packing cartons for Hami melons (40 cm × 35 cm × 28 cm) in four boxes each, respectively, and then transported outside for cold storage at 3 °C.

The reagents *n*-butanol, glacial acetic acid, choline oxidase, horseradish peroxidase, phenol, 4-aminoantipyrine, dichloromethane, and isooctane were all analytically pure. The reagents methanol, acetonitrile, sodium chloride, isooctane, isopropanol, formic acid, and hexane were chromatographically pure. PI and PA were purchased from Sigma-Aldrich (St. Louis, MO, USA), and the fatty acids were purchased from Shanghai Ampoule (Shanghai, China).

### 2.2. Instruments and Equipment

The main instruments of the experiment and their related information are detailed in Table 1. 

### 2.3. Methods

#### 2.3.1. Sample Handling

The samples were divided into two groups, and the Hami melons were soaked in water as the control and 1.0% *n*-butanol for 30 min, which was the optimal concentration of *n*-butanol, and then placed in cold storage at 3 °C for low-temperature storage as described by Ting et al. [19]. The samples were taken once every 7 d for seven times, and melon rinds that were approximately 2 cm thick were cut around the equator of Hami melon fruits, chopped and mixed quickly, and then frozen in liquid nitrogen and immediately stored at −40 °C.

#### 2.3.2. Determination of the Chilling Injury Index

Fruits to be tested were transferred from cold storage to room temperature (25 °C) for 2 d and then analyzed as described by Bi et al. [20].
$$\mathrm{Cold}\;\mathrm{damage}\;\mathrm{index}=\sum \frac{\mathrm{Fruit}\;\mathrm{cold}\;\mathrm{damage}\;\mathrm{level}\times \mathrm{Number}\;\mathrm{of}\;\mathrm{fruits}\;\mathrm{at}\;\mathrm{this}\;\mathrm{level}}{\mathrm{Total}\;\mathrm{number}\;\mathrm{of}\;\mathrm{fruits}\times \mathrm{Highest}\;\mathrm{level}}$$

#### 2.3.3. Determination of Cell Membrane Permeability

The method of Zhang Ting et al. [21] was used for determination. Fifteen fruit slices were taken along the equatorial plane of the fruit by using a stainless steel drill with a diameter of 10 mm, placed in a small beaker of 50 mL, 40 mL distilled water was added, and the mixture was stirred slowly for 5 s. The conductivity was measured by a conductivity meter, and then the conductivity *P*_1_ was measured after 10 min. Then the small beaker was placed in a boiling water bath for 10 min, and 40 mL of distilled water was added. The electrical conductivity *P*_2_ was determined. The assay was repeated three times and the average value was taken. The calculation formula is as follows:$$\mathrm{cell}\;\mathrm{membrane}\;\mathrm{permeability}\left(\%\right)=\frac{\left(P_{1}-P_{0}\right)}{\left(P_{2}-P_{0}\right)}\times 100$$

#### 2.3.4. Determination of MDA Content

As described by Cao et al. [22], after weighing 1.0 g of fruit and vegetable samples, 5.0 mL of 100 g/L trichloroacetic acid solution was added followed by grinding and homogenization. The supernatant was collected by centrifugation at 10,000× *g* for 20 min at 4 °C and stored at a low temperature for later use.

Next, 2.0 mL of supernatant (2.0 mL of 100 g/L thiobarbital acid solution was added to the control blank tube instead of the extract) was mixed with 2.0 mL of 0.67%TBA and boiled in a boiling water bath for 20 min. Then it was removed and cooled and centrifuged again. The absorbance values of the supernatant at 450 nm, 532 nm, and 600 nm wavelengths were determined. Repeat three times.

According to the absorbance value of the solution, the content of *MDA* in the reaction mixture was calculated according to the previous formula, and then the content of *MDA* in each gram of fruit and vegetable sample (fresh weight) was calculated according to the following formula, which was expressed as μmol/gm_F_. The calculation formula is as follows:$$MDA\;content=\frac{c\times V}{V_{s}\times m\times 1000}\left(\mu \mathrm{mol}/{\mathrm{gm}}_{F}\right)$$
where, *c*—malondialdehyde concentration in reaction mixture, μmol/L;

*V*—Total volume of sample extract, mL;

*Vs*—the volume of sample extraction liquid taken during the determination, mL; 

*m*—sample mass, g.

#### 2.3.5. Determination of the Contents of PI and PA

The content of PI and PA was determined according to the method of Xu et al. [23], a total of 0.1 g of Hami melon rind was weighed, and 200 μL of H2O was then added. The solution was vortexed and mixed well. A volume of 400 μL of methyl tert-butyl ether (MTBE) and 80 μL of MeOH was added and then vortexed. The extract was treated with ultrasound for 30 min and then dissolved in acetonitrile: isopropanol at high speed. The contents were measured using ultra-performance liquid chromatography coupled with mass spectrometry (UHPLC-QE).

Chromatographic column conditions: Waters HSS T3 (100 × 2.1 mm, 1.8 μm) (Waters, Milford, MA, USA). The mobile phase was acetonitrile: water (2:8, *v*/*v*) for phase A and acetonitrile: isopropanol (2:8, *v*/*v*) for phase B; flow rate 0.3 mL/min; column temperature 40 °C; injection volume 2 μL; elution gradient: 0 min A/B (70:30 *v*/*v*), 4 min A/B (70:30 *v*/*v*), 22 min A/B (0:100 *v*/*v*), 22.1 min A/B (70:30 *v*/*v*), and 26 min A/B (70:30 *v*/*v*).

Mass spectrometry conditions: The electrospray ionization (ESI) conditions were as follows: sheath gas 60 arb; auxiliary gas 20 arb; ion spray voltage 3000 V; temperature 285 °C; and ion transport tube temperature 370 °C. The scanning mode was full scan detection (FSMS2) mode and a positive ion. The primary scan range (scan m/z range) was 80–1200.

#### 2.3.6. Determination of the Contents of Fatty Acids

The fatty acid determination was determined according to the method of Ge et al. [24], a volume of 100 mg of frozen sample was placed in an EP tube, and 4 mL of chloroform was added. The solution was vortexed for 30 s and mixed well. It was centrifuged at 3500 rpm for 15 min at room temperature, removed, and left standing. The lower phase was removed with an automatic pipette, transferred to another tube, and 2 mL of dichloromethane was added. The mixture was vortexed for 30 s and centrifuged for 15 min. The lower phases were combined and dried with a stream of nitrogen. After derivatization, the sample was placed in the vial to be measured and analyzed using gas chromatography coupled with mass spectrometry (GC-MS).

Chromatographic conditions: gas chromatography (Agilent 6890; Agilent Technologies, Santa Clara, CA, USA) CP-Sil 88 (100 m × 0.25 mm × 0.25 µm) column, ramp-up procedure: held at 100 °C for 5 min, ramped up to 240 °C at 4 °C/min and held for 15 min; carrier gas (He) at a flow rate of 1.0 mL/min; injection volume of 1 μL; and a splitting ratio of 10:1.

Mass spectrometry conditions: quadrupole mass spectrometry detection system (Agilent 5977; Agilent Technologies), inlet temperature 260 °C, quadrupole temperature 150 °C; scanning mode was full scan mode (SCAN), mass scan range m/z 30~550.

#### 2.3.7. Assay of LOX

The LOX activity was assayed as described by Malekzadeh et al. [25].

(1)Preparation of reaction substrate: The enzyme reaction substrate used in the test was 10 mmol/L sodium linoleate, and the amount of 70 mg sodium linoleate, 70 mLTriton X-100, and 4 mL anaerobic water were mixed (to avoid bubbles). After titration with 0.5 mol/L sodium hydroxide, the solution was clarified, the volume was 25 mL, and the volume was divided into 1.0–1.5 mL and stored at −18 °C until use.(2)Extraction of crude enzyme solution: 2.0 g of pulp tissue was placed in a mortar, ground with liquid nitrogen, added with 10 mL of 50 mmol/L (pH 7.0) phosphate buffer precooled at 4 °C, centrifuged at 15,000× *g* (4 °C) for 15 min, and the supernatant was used for LOX activity determination.(3)Determination of enzyme activity: the reaction system of 3 mL contained 25 μL sodium linoleate mother liquor, 2.775 mL buffer, and 0.2 mL enzyme solution, and the reaction temperature was 30 °C. The LOX activity was measured at 234 nm. The timing began 15 s after adding the enzyme solution, and the OD value change was recorded within 1 min. The enzyme activity was measured as ∆OD_234_/gF_W_·min. The procedure was repeated 3 times.

#### 2.3.8. Assay of PLD

The PLD was assayed as described by Aghdam et al. [26], The 50 g sample was homogenized in 300 mL of water at 4 °C for 5 min, filtered through gauze, and centrifuged at 13,000× *g* 4 °C for 30 min, and the separated supernatant after centrifugation was used as PLD crude enzyme extract. PLD enzyme activity was measured according to the release of choline from PLD hydrolysis of the substrate PC. For the composition of a substrate PC solution, 40 gPC was dissolved in 1 mL of chloroform and 5 mL of water containing 1.44 mg/mL of sodium dodecyl sulfate (SDS). For the reaction system, 10 μL PC, and 30 μL 0.5 mol/L sodium acetate buffer (pH 5.5) containing 5 mmol/L CaCl_2_ were added to 30 μL crude enzyme extract in a water bath at 37 °C for 10 min. The reaction was terminated by adding 20 μL 1 mol/L Tris-HCl buffer (pH 8.0) containing 0.1 mol/L EDTA. Then, 30 uL (containing 0.2 U horseradish peroxidase, 1.5 mmol/L 4-aminoantipyrin, 2.1 mmol/L phenol, 3U choline oxidase, and 10 mmol/L Tris-HC1 buffer pH 8.0) was quickly added to a 37 °C water bath for 45 min. This reaction solution was colorimetric at 492 nm and the change in the A492 value was recorded over 3 min. The activity of PLD was calculated as one unit of activity per two pairs required to increase the absorbance value of the reaction system at a wavelength of 492 nm by 0.01 per minute. The unit is 0.01ΔOD_492_/min·mg protein. The protein content in the enzyme extract was determined according to the Coomassie brilliant blue method.

#### 2.3.9. RNA Extraction, cDNA Synthesis, and Primer Analysis by Quantitative Fluorescent PCR (Q-PCR)

Samples of Hami melon stored at 3 °C for 14, 28, and 42 d were subjected to fluorescent quantitative PCR using the 2-ΔΔCt analytical method. The primer sequences used are shown in Table 2.

#### 2.3.10. Gene Cloning

The analysis of the pre-transcriptome indicated that the PLD-β gene was significantly more differentially expressed than the other types, and the gene was subsequently cloned using the homologous cloning method combined with a 3′-RACE end rapid amplification method. Information of primers used in this gene cloning used is shown in Table 3. 

#### 2.3.11. Data Processing and Analysis

After processing, samples were taken every 6 days, totaling 7 times. We took 3 melons from each treatment, performing 3 replicates for a total of 9 melons, and repeated the measurement three times for each indicator.

The experimental data were analyzed by Microsoft Excel 2010 (Redmond, WA, USA) and plotted using Origin9 (OriginLab, Northampton, MA, USA). The data were statistically analyzed and their significance was determined as *p* < 0.05 using SPSS 17.0 (SPSS, Inc., Chicago, IL, USA). A bioinformatics analysis was performed on *CmPLD-β* using BLASTP on the NCBI website to identify the open reading frame of the gene sequence studies and analyze its conserved structural domains and homology of nucleic acids for translation and multiple sequence comparison. The evolutionary trees were systematically constructed and analyzed using DNAMAN 7.0, as well as ClustalX2.0.11.

## 3. Results

### 3.1. Effects of n-Butanol Treatment on Chilling Injury Symptoms and Chilling Injury Index of Hami Melon Fruits during Storage

As shown in Figure 1, chilling injury occurred in the Hami melons in both treatments after 14 d of refrigeration and 2 d at room temperature, and the chilling injury index and the symptoms of chilling injury increased with time. There were significant differences in chilling injury symptoms at the cold stage of 28 d and 42 d of storage at room temperature, and *n*-butanol treatment significantly reduced the symptoms of chilling injury in Hami melon fruits (Figure 2). The chilling injury index of Hami melon in the control group was higher than that in the group treated with *n*-butanol at 35–42 d. At 42 d of storage, the chilling injury index of the control group was 44% and that of the group treated with *n*-butanol was 33% with a significant inhibitory effect (*p* < 0.05). This indicated that *n*-butanol significantly inhibited the increase of chilling injury index in Hami melon fruits at the late stage of storage (*p* < 0.05).

### 3.2. Effect of n-Butanol on Cell Membrane Permeability and the MDA Content of Hami Melon

As shown in Figure 3a, the cell membrane permeability of the pericarp gradually increased with the extension of storage time. The cell membrane permeability of the group treated with *n*-butanol was always lower than that of the control at the same time of storage. During 21–42 d of storage, the cell membrane permeability of the group treated with *n*-butanol was 28.2%, 12.08%, 27.77%, and 11.19% lower than that of the control group, respectively (*p* < 0.05). The results indicated that treatment with *n*-butanol could effectively inhibit the enhancement of cell membrane permeability of Hami melon, and the effect was significant in the late storage period (*p* < 0.05).

Figure 3b shows an overall increase in the MDA content of Hami melon, and both treatments showed similar trends. An additional comparison showed that the MDA content of the group treated with *n*-butanol was lower than that of the control group at the same storage time (except 14 d), and the MDA content of the group treated with *n*-butanol was significantly lower than that of the control group at 7, 21, and 42 d (*p* < 0.05). The results indicated that treatment with *n*-butanol could effectively alleviate the increase in MDA content in Hami melon fruits during storage.

### 3.3. Effect of n-Butanol on the PI and PA Contents of Hami Melon

As shown in Figure 4a, the PI content of Hami melon increased overall during storage, and both treatments showed the same trend with fluctuating increases. Further comparison showed that the PI content of the group treated with *n*-butanol was higher than that of the control during the same storage period, and the PI content of the group treated with *n*-butanol was higher than that of the control group by 18.59%, 81.94%, and 22.1% at d 7, 21, and 42, respectively. There was a significant difference between them (*p* < 0.05). The overall PA content of the Hami melon fruit fluctuated upward (Figure 4b), and the control group clearly fluctuated more with a sharp increase and decrease from 0 to 21 d of storage, respectively, and a sharp increase and decrease occurred again from 21 to 35 d of storage, respectively. The content then increased to the end of storage, while the *n*-butanol treatment increased slowly throughout the storage period. An additional comparison revealed that at the same storage time, the PA content of the group treated with *n*-butanol was lower than that of the control group (except for 21 d) and was 34.3% and 46.85% lower than that of the control group at 14 and 28 d of storage, respectively (*p* < 0.01). These results indicate that treatment with *n*-butanol can effectively enhance the content of PI and reduce that of PA.

### 3.4. Effect of n-Butanol on Saturated Fatty Acid Content of Hami Melon

Two saturated fatty acids were measured in the Hami melon rind, including stearic acid (C18:0) and palmitic acid (C16:0). As shown in Figure 5a, the stearic acid content in the control group decreased rapidly from 0–7 d, increased rapidly to its highest point from 7–21 d, and continued to decrease to the lowest point of 0.118 μg·mg^−1^ from 21 d to the end of storage. In contrast, the group treated with *n*-butanol decreased slowly from 0–21 d, increased rapidly from 21–28 d, and decreased slowly from 28–42 d. An additional comparison revealed that at the same storage time, the stearic acid content in the group treated with *n*-butanol was lower than that in the control group at 0–21 d, particularly 18.12% and 34.13% μg·mg^−1^ at 14 d and 21 d, respectively (*p* < 0.01). The stearic acid content in the group treated with *n*-butanol was higher than that in the control group at 28–42 d of storage.

As shown in Figure 5b, the palmitic acid content of both treatments showed large fluctuations up and down. The control group decreased rapidly from 0–7 d, increased rapidly at 7–14 d, and decreased rapidly followed by a rapid increase at 14–28 d, and a continuous slow decrease occurred from 28 d to the end of storage. The palmitic acid content in the group treated with *n*-butanol fluctuated more than that in the control group, reaching a peak of 0.33 μg·mg^−1^ after a rapid “decrease–increase” from 0–21 d of storage, at which time the palmitic acid content was 23.55% higher than that in the control group (*p* < 0.01) and began to decrease at 21 d of storage until the end of storage. The palmitic acid content started to decrease at 21 d of storage until the end of storage. Thus, treatment with *n*-butanol had little effect on the content of palmitic acid in Hami melon fruits.

### 3.5. Effect of n-Butanol on the Unsaturated Fatty Acid Content of Hami Melon

A total of four unsaturated fatty acids were measured in the Hami melon rind, including linoleic acid (C18:2), linolenic acid (C18:3), oleic acid (C18:1), and erucic acid (C22:1n9).

As shown in Figure 6a, the trends of linoleic acid content were approximately the same in both treatments, and the linoleic acid content in the group treated with *n*-butanol was higher than that in the control group throughout the storage period (except 42 d), and the linoleic acid content in the group treated with *n*-butanol was significantly higher than that in the control group at storage times of 7, 28, and 35 d by 11.88%, 10.04%, and 20.45% μg·mg^−1^, respectively (*p* < 0.05). The results indicated that treatment with *n*-butanol was more effective at increasing the content of linoleic acid (C18:2) and unsaturated fatty acid.

The overall trend of the oleic acid content of Hami melon fruit decreased under low-temperature storage (Figure 6b). An additional comparative analysis showed that the oleic acid content of Hami melon fruit in the group treated with *n*-butanol was significantly higher than that of the control group at 7, 28, and 35 d of storage by 24.75%, 15.68%, and 22.31% μg·mg^−1^, respectively (*p* < 0.05). This indicates that *n*-butanol was more effective at enhancing the content of oleic acid in Hami melon in the later stages of storage.

As shown in Figure 6c, the linolenic acid content of Hami melon fruits fluctuated during storage and both the fruit treated with *n*-butanol and the control showed the same trend. However, the linolenic acid content of the group treated with *n*-butanol was higher than that of the control group throughout the storage process (except 42 d) and was significantly elevated in fruits at 7, 28, and 35 d of storage by 18.6%, 10.22%, and 15.96% μg·mg^−1^, respectively (*p* < 0.05). The results indicated that *n*-butanol helped to alleviate the decrease in the linolenic acid content of Hami melon fruits.

The erucic acid content in the control group fluctuated at 7 d intervals and in opposite directions, with two “decreases and increases” followed by a continuous decrease to the lowest value of 0.069 μg·mg^−1^ at 35–42 d (Figure 6d). The group treated with *n*-butanol showed a more moderate trend, and a more detailed comparison revealed that the erucic acid content in the group treated with *n*-butanol was significantly higher than that of the control group at 28 and 42 d of storage by 13.6% and 19.34%, respectively at the same storage time (*p* < 0.01). The effect of treatment with *n*-butanol on the erucic acid content of Hami melon was more significant during the late storage period. These results indicated that *n*-butanol could increase the contents of linoleic acid, linolenic acid, and oleic and erucic acid in Hami melon at the late storage stage (28–42 d).

### 3.6. Effect of n-Butanol on LOX Activity and CmLOX Expression in Hami Melon

As shown in Figure 7a, LOX activity increased overall during storage, and at the same storage time, the LOX activity in the control group was higher than that in the group treated with *n*-butanol, particularly during the late storage period (28–42 d). The control group was 228.66%, 113.83%, and 88.98% U·mg^−1^ higher than that in the group treated with *n*-butanol, respectively (*p* < 0.01). The level of expression of CmLOX was lower in the control group than in the group treated with *n*-butanol at 14 and 28 d of storage in contrast to LOX activity and higher than in the group treated with *n*-butanol at 42 d in contrast to LOX activity (Figure 7b). This showed that treatment with *n*-butanol significantly reduced the LOX activity and level of expression of CmLOX at 42 d of storage (*p* < 0.05).

### 3.7. Effect of n-Butanol on PLD Activity and CmPLD-β Expression in Hami Melon

As shown in Figure 8a, the overall PLD activity in both treatment groups fluctuated upward with the same trend in both treatments, and the PLD activity in the *n*-butanol treatment group was lower than that in the control group throughout the storage period. The trend of *n*-butanol treatment was the same as that of the control group during the early stage, but it started to decrease after a brief increase from 21 to 28 d until the end of storage. A more detailed comparison revealed that the group treated with *n*-butanol was 32.81%, 47.67%, and 29.25% lower than that of the control group at d 7, 14, and 35 d of storage, respectively, which was highly significant (*p* < 0.01), and Figure 8b shows that the level of expression of *CmPLD-β* in the group treated with *n*-butanol was lower than that in the control group at the selected storage time points, which was consistent with the PLD activity (Figure 8a). The differences were 2.17- and 7.95-fold higher in the group treated with *n*-butanol at d 14 and 42, respectively (*p* < 0.05). Thus, it can be concluded that treatment with *n*-butanol reduced the PLD activity and level of expression of *CmPLD-β*.

### 3.8. Sequence Analysis of the Hami Melon PLD-β Gene

The sequencing results were compared with those on the NCBI website using BLASTX, and the fragment was found to be 100% homologous to PREDICTED: phospholipase D beta 2-like [Cucumis melo] (XP_008460150). The sequencing results showed that *CmPLD-β* encodes 1103 amino acids, and the molecular weight of the encoded protein is 123.707 kDa with an isoelectric point of 7.06.

#### 3.8.1. Evolutionary Tree Analysis of the Base Sequence Homology of Hami Melon Rind PLD-β Gene with Multiple Species

Phylogenetic tree nucleotide alignment of *PLD-β* from various plant species and Hami melon per-icarp is shown in Figure 9.

#### 3.8.2. Evolutionary Tree Analysis of Nucleic Acid Sequence Homology of PLD-β Gene in Hami Melon Pericarp with Multiple Species

Phylogenetic tree of the amino acid sequence alignment of *CmPLD-β* from various plant species and Hami melon pericarp is shown in Figure 10. 

#### 3.8.3. Twofold Comparison of the CmPLD-β Gene and Target Gene in Hami Melon Pericarp

Double comparison of PLD-β from the target gene and Hami melon pericarp is shown in Figure 11. 

## 4. Discussion

Low-temperature storage is the primary postharvest storage method to maintain fruit quality and commercial value [27]. Refrigeration of Hami melon fruits at 3 ± 0.5 °C helps to maintain fruit quality and prolong the postharvest storage and preservation period of Hami melon [19]. However, Hami melon is sensitive to the cold, and after long-term low-temperature storage, the fruit skin is prone to deteriorate, particularly when the fruit is moved from refrigeration to room temperature where chilling injury symptoms become more apparent. The severity of deterioration positively correlated with the duration of refrigeration. In this study, sunken spots appeared on the epidermis of Hami melon after low-temperature refrigeration, which was similar to the results of Wang et al. [28], and treatment with *n*-butanol significantly reduced the chilling injury index of fruits in the late storage period. Thus, it was effective at alleviating the symptoms of chilling injury (Figure 1 and Figure 2). The contents of MDA and cell membrane permeability reflected the degree of damage to the epidermal cell membrane, and both MDA and the cell membrane permeability of Hami melon fruits under low-temperature stress increased significantly. *n*-butanol could inhibit the increase in MDA content and cell membrane permeability, which is consistent with the results of Wang et al. [28]. This indicates that treatment with *n*-butanol can alleviate the rate of membrane lipid degradation in Hami melon pericarp and maintain the structure and function of the membrane, thus reducing chilling injury symptoms and the chilling injury index.

Studies have shown that unsaturated fatty acids can stabilize the degradation of membrane lipids by serving as enzymatic substrates [29]. In this study, an overall analysis of cell membrane lipids revealed a large proportion of two saturated fatty acids (stearic acid and palmitic acid) and four unsaturated fats (linoleic acid, linolenic acid, oleic acid, and erucic acid) in Hami melon fruits. However, during the 35–42 storage time, the unsaturated fatty acids(18:0) and (16:0) in treated and control fruits showed sharp decreases and increases, and then reached the minimum and max values at 42 d (Figure 5a,b); we determined that with the development of cold storage, the linoleic acid (C18:2), linolenic acid (C18:3), and oleic acid (C18:1) (Figure 6a–c) showed similar trends, indicating the membrane lipid metabolism balance. However, on the 28th day, the content of PA in the treated fruits was higher than those of the control. This trend may be related to the gene expression of PLD, with almost no difference in PLD gene expression between the control and treatment on day 28 (Figure 8b). Oleic acid in the treated fruits was higher than those of the control, while the erucic acid and oleic acid contents were lower than that of the control on day 28. The reason was the lack of difference in gene expression between LOX-treated and control groups (Figure 7b). These phenomena were consistent with the gene expression of PLD and LOX. However, no such conclusions were found in other studies.

Under low-temperature stress, the unsaturated fatty acid content of the substrate was reduced owing to the activation of LOX, and the cell membrane was damaged, which resulted in chilling injury to the fruit. PI is an important phospholipid component that can activate antioxidant enzymes and reduce the signaling molecules of free radicals that are toxic to plant cells, and PA is a direct product of PLD hydrolysis of the cell membrane. It is apparent that treatment with *n*-butanol more effectively reduced the PA content and increased the level of PI, which is partly the same as the results of a study on the membrane lipid fractions in apple (*Malus domestica*) fruits [30], indicating that the membrane lipid fractions respond to low-temperature stress and are one of the causes of fruit damage.

PLD and LOX, the key enzymes of membrane lipid metabolism, primarily disrupt the cell membranes by degrading fatty acids or phospholipids in the epidermal membrane, which causes abnormal cell metabolism and, thus, reduces cellular functions [12]. Cold stress activates PLD and LOX activities in Hami melon fruits, and a study of the levels of expression of CmPLD and CmLOX showed that the level of expression of CmPLD was significantly downregulated in the group treated with *n*-butanol of Hami melon, which is consistent with a significant reduction in PLD activity by *n*-butanol, indicating that treatment with *n*-butanol negatively regulates PLD activity and gene expression in fruits under cold stress, which is closely related to the β-type of the PLD gene family (transcriptome). The results showed that the level of expression of *CmPLD-β* increased more at 42 d of cold storage, presumably owing to the disruption of cell membrane structure caused by prolonged low temperature, increased membrane permeability, and the degradation of membrane lipids, which is consistent with the chilling injury index and chilling injury symptoms (Figure 1 and Figure 2). In addition, LOX activity was more effectively suppressed under *n*-butanol treatment. However, at the same time points of 14 d and 28 d after treatment with *n*-butanol, the level of expression of CmLOX increased, which is consistent with research on blueberry (*Vaccinium* sp.) [31,32]. This could be owing to the fact that there are different members of the LOX gene family and differences in the transcript levels of varied gene members in abiotic stresses.

The PLD-β gene of the Hami melon rind was cloned, and the amino acid sequence of *CmPLD-β* obtained from Hami melon fruit was deduced by a BLASTP comparison on the NCBI website, and the predicted amino acid sequence was sequenced with those of other 29 species to construct a phylogenetic tree. It is apparent that the Hami melon *CmPLD-β* is more closely related to cucumber (CsXP5) with 100% homology and is 87% homologous with that of pumpkin (*Cucurbita* spp.) (CmXP1). This indicates that closer homology indicates a more similar function. The base sequences of *CmPLD-β* and CmXP5 were analyzed by a twofold comparison, and the promoter region of the previous segment of CmPLD was amplified experimentally. The base sequences of both promoter regions were inconsistent, indicating that the transcription factors that regulated the promoters vary between two fruits of the Cucurbitaceae family with different varieties.

## 5. Conclusions

Treatment with *n*-butanol reduced the pericarp MDA content, cell membrane permeability, PLD and LOX activities, and the levels of expression of *CmPLD-β* and *CmLOX* at 42 d. In addition, this treatment increased the production of membrane lipid fraction PI content and the unsaturated fatty acids linoleic acid, linolenic acid, and oleic and erucic acid contents during the late storage period (28–42 d), and inhibited PA production and the saturated fatty acid stearic acid, which alleviated the chilling injury of fruit to some extent. The cloning of the PLD-β gene from the Hami melon rind showed that the *CmPLD-β* of the Hami melon is closely related to that of cucumber (CsXP5) with 100% homology, indicating that PLD in Hami melon fruit is functionally identical to PLD in cucumber fruit.

## Figures and Tables

**Figure 1 foods-12-01904-f001:**
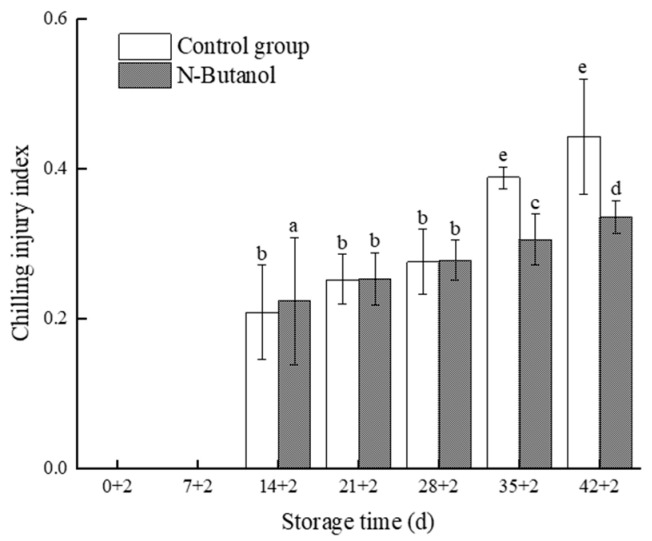
Effect of *n*-butanol on the chilling injury index of Hami melon during cold storage. Different lowercase letters indicate significant differences among different treatments at the same storage time (*p* < 0.05).

**Figure 2 foods-12-01904-f002:**
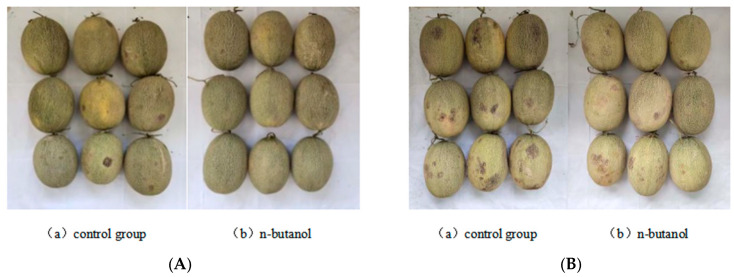
Symptom of chilling injury of Hami melon fruits after cold storage (**A**) for 28 d (**a**) and 42 d (**b**) at room temperature for 2 d (**B**).

**Figure 3 foods-12-01904-f003:**
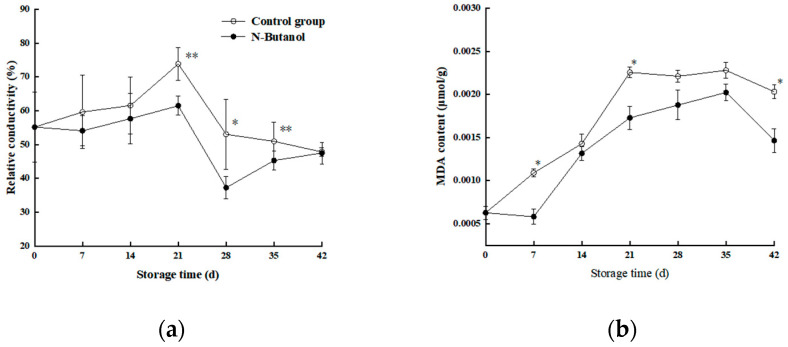
Effect of *n*-butanol on electrolyte leakage (**a**) and MDA content (**b**) of Hami melon fruits. MDA, malondialdehyde. The asterisks indicate that the values are significantly different between groups at the same time point (* *p* < 0.05, ** *p* < 0.01).

**Figure 4 foods-12-01904-f004:**
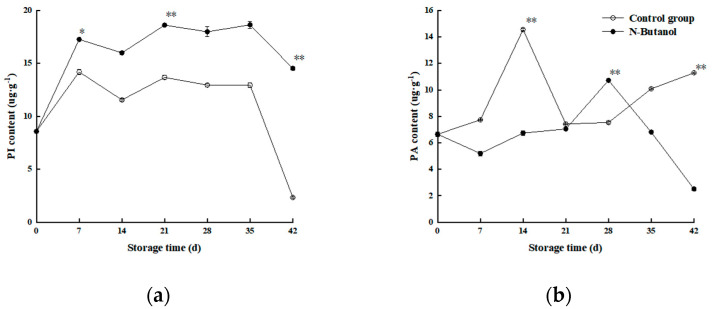
Effect of *n*-butanol on the contents of PI (**a**) and PA (**b**) of Hami melon fruits. The asterisks indicate that the values are significantly different between groups at the same time point (* *p* < 0.05, ** *p* < 0.01).

**Figure 5 foods-12-01904-f005:**
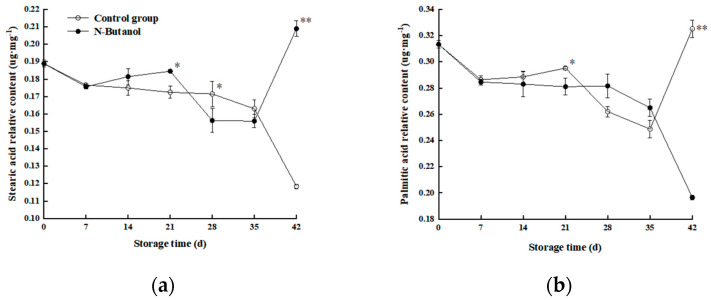
Effect of *n*-butanol on stearic acid (C18:0) (**a**) and palmitic acid (C16:0) (**b**) relative content in Hami melon fruits. The asterisks indicate that the values are significantly different between groups at the same time point (* *p* < 0.05, ** *p* < 0.01).

**Figure 6 foods-12-01904-f006:**
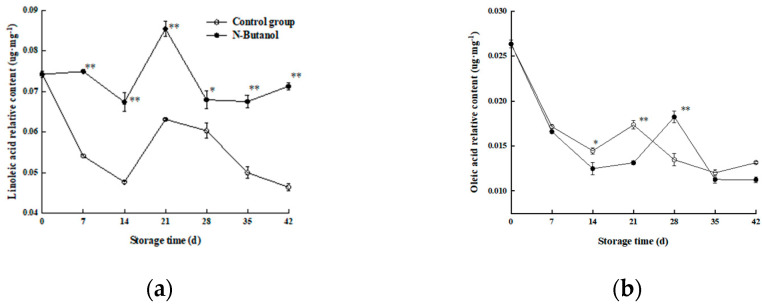
Effect of *n*-butanol on linoleic acid (C18:2) (**a**), oleic acid (C18:1) (**b**), linolenic acid (C18:3) (**c**), and erucic acid (C22:1n9) (**d**) relative contents in Hami melon fruits. The asterisks indicate that the values are significantly different between groups at the same time point (* *p* < 0.05, ** *p* < 0.01).

**Figure 7 foods-12-01904-f007:**
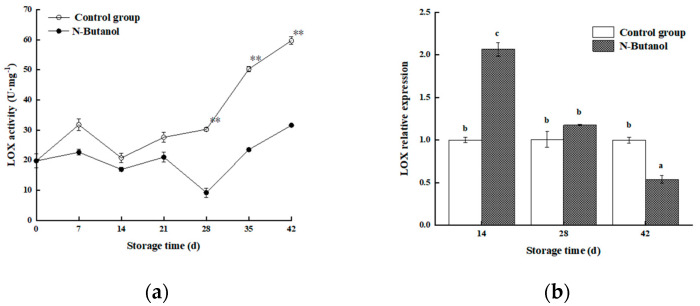
Effect of *n*-butanol on the LOX (**a**) and expression of CmLOX (**b**) in Hami melon fruits. The asterisks indicate that the values are significantly different between groups at the same time point (** *p* < 0.01). Different lowercase letters indicate significant differences among different treatments at the same storage time (*p* < 0.05).

**Figure 8 foods-12-01904-f008:**
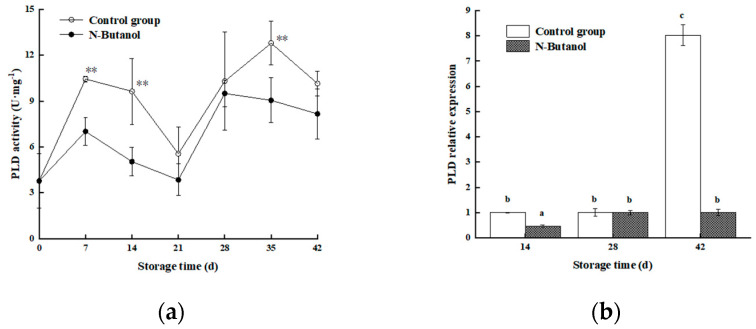
Effect of *n*-butanol on the PLD (**a**) and expression of *CmPLDβ* (**b**) in Hami melon fruits. The asterisks indicate that the values are significantly different between groups at the same time point (** *p* < 0.01). Different lowercase letters indicate significant differences among different treatments at the same storage time (*p* < 0.05).

**Figure 9 foods-12-01904-f009:**
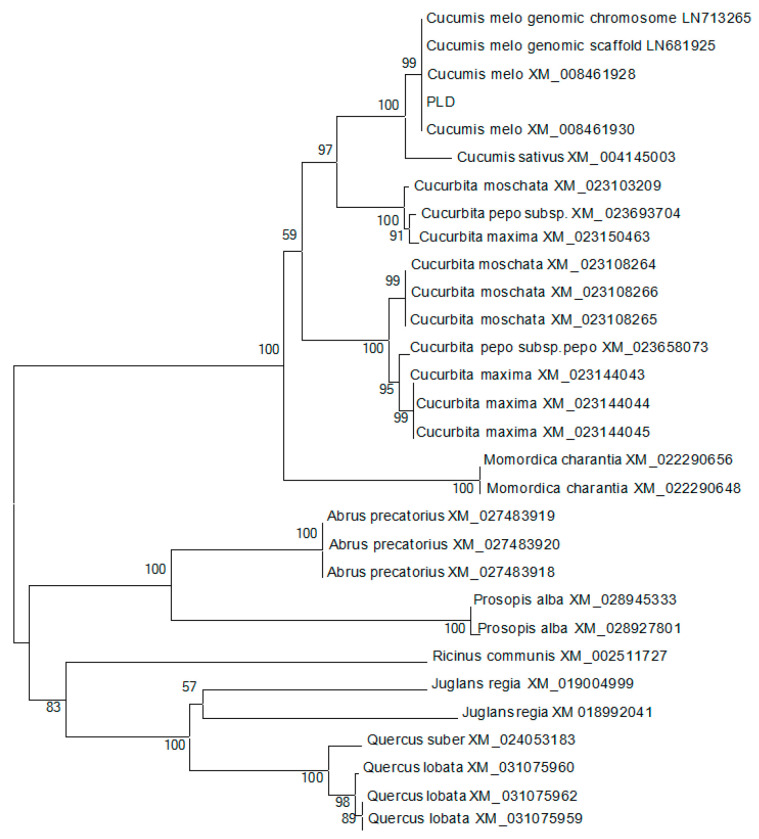
Phylogenetic tree nucleotide alignment of PLD-β from various plant species and Hami melon pericarp.

**Figure 10 foods-12-01904-f010:**
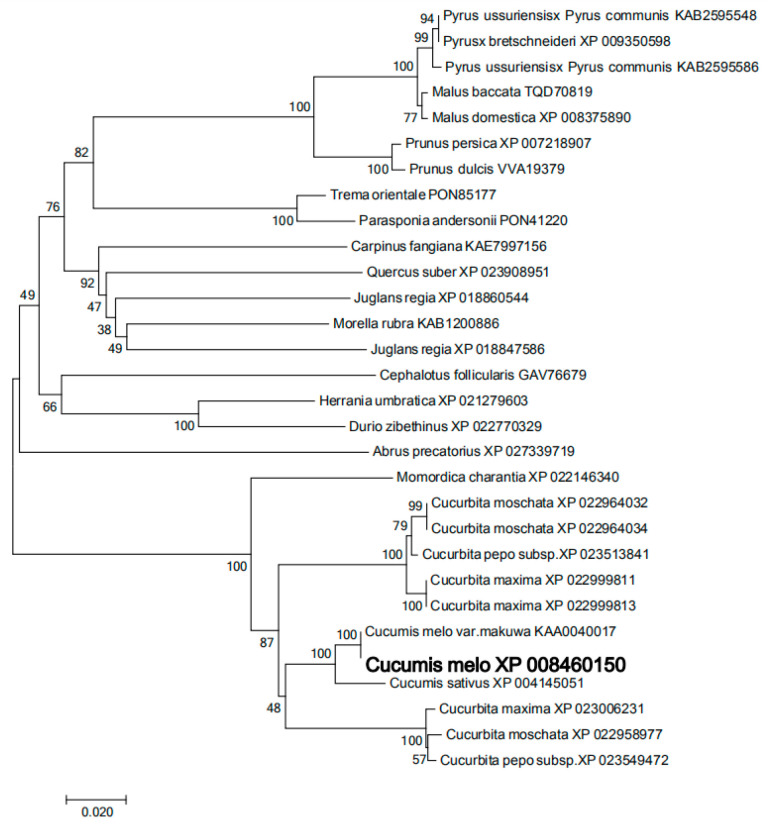
Phylogenetic tree of the amino acid sequence alignment of *CmPLD-β* from various plant species and Hami melon pericarp.

**Figure 11 foods-12-01904-f011:**
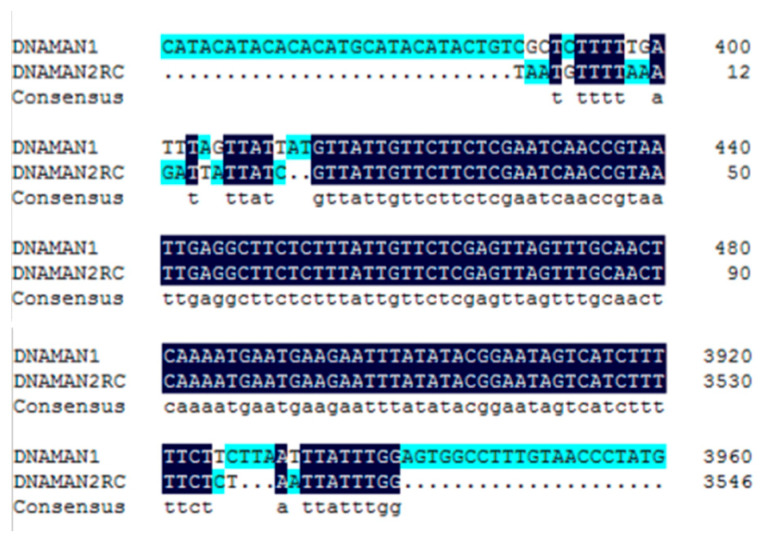
Double comparison of PLD-β from the target gene and Hami melon pericarp.

**Table 1 foods-12-01904-t001:** The main test instruments.

Instrument Name	Instrument Model	Instrument Manufacturers
Nitrogen purge instrument	Thermo	Reacti-thermo (MA, USA)
Mass Spectrometer		Q Exactive (MA, USA)
UPLC		Vanquish (MA, USA)
Freeze dryer	Four rings	LGJ-10 (Beijing, China)
Enzyme Markers	HBS-1101	Nanjing Detie Test Equipment Co. (Nanjing, China)

**Table 2 foods-12-01904-t002:** The primer sequence of qRT-PCR.

Category	Accession Number	Sequence (5′ to 3′)
CmPLD	CmPLD-F	CAGGCAGAGAATGAGAACAACA
	CmPLD-R	AGGGGATATGTGGATAATCCGT
CmLOX	CmLOX-F	GCACAACACGCAGCACTAAA
	CmLOX-R	CTCTGGA TCGTTCTCGTCGG
18S	18S-F	GCTGTCACTGTTTTTGGCGT
	18S-R	GCACCACCCTTCAAATGAGC

**Table 3 foods-12-01904-t003:** Information of primers used in this gene cloning.

Primer Name	Primer Sequences
PLD-F	ACCCATACCCTCGTCCAATTCCATCTC
PLD-R	TGTCCGTCTGGAACATGGGCATCTTG
PLD-F	AGCCATGTAAGCCTGGAGCTGCTCTAAG
PLD-R	TGAACTCCATAGGGTTACAAAGGCCACTC
PLD-F	TCCATATCACAATCCTTACCCATACCCTC
PLD-R	AGATAGTGAGGTTCTCCTGAATTCCAAG
PLD-F	ACACACATGCATACATACTGTCGCTC
PLD-R	ACTCCATAGGGTTACAAAGGCCACTC

## Data Availability

Data is contained within the article.

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
