# Peer review of "Reduction of Membrane Lipid Metabolism in Postharvest Hami Melon Fruits by n-Butanol to Mitigate Chilling Injury and the Cloning of Phospholipase D-β Gene"

_foods, 2023, doi:10.3390/foods12091904_

Round 1

Reviewer 1 Report

Title:

 Reduction of membrane lipid metabolism in postharvest Hami melon fruits by n-butanol to mitigate chilling injury and the 3 cloning of phospholipase D-β gene

This research the effect of n-butanol on the postharvest membrane lipid metabolism of Hami melon fruits and cloned the CmPLD-β was investigated. The writing and the research design are good, but there are some ambiguities in the text that need to be corrected.

All the scientific name through the text should be in italic form.

Line 62-64 reference?

In introduction section state the PLD is the abbreviation for?

Lines 200-207 are referring to statistical analysis and should be insert at the end of material and method section, not the beginning of result, move it. Also, some detail should add to this section: the experimental design? The replication numbers? The mean comparison test name? The test for checking normality?

Section 2.3.2. state some detail on the ranking score, what was the minimum rank and what was the maximum rank.

Section 23.5. and 2.3.6 add the reference

For all the chemical agent check the writing format: in some case the numbers should be in superscript format (line 147, 190) and in some cases in subscript form (line 147), revise please.

Line 208 the title is not accepted revise it.

Chilling injury is a qualitative trait so there is no need for doing mean comparison for this trait. Delete the letter of significant from the columns of figure 2.

Figure 3a: the SD for control group is high and the difference between the replications is very high. re-check the data please.

Figure 5a and 5b: state the reason for the sharp decrease and increase in day 35 in discussion section.  

Line 287 and 327, why relative content? The unit for the traits is not percentage, why the title of figure contains relative?

Line 314-318 the sentence is too long and unclear, revise.

Figure 8a: the SD for control group is high and the difference between the replication is very high. re-check the data please.

There are no explains for section 3.8.1, 3.8.2., 3.8.3 please revise.

Line 475 and 493 check the format for the name, all the alphabets are presented in capital form.

Author Response

感谢审稿人的评论,我们将在下面的文章中详细回复您的评论。

Reviewer 2 Report

The article titled " Reduction of membrane lipid metabolism in postharvest Hami melon fruits by n-butanol to mitigate chilling injury and the cloning of phospholipase D-β gene" is pertinent to the Journal. The subject of treatments for postharvest food preservation is critical and topical now. However, its results must be better argued and analyzed.

Here are some suggestions:

Introduction

All Scientifical names must be in the italic letter.

Material and methods

Please, it is necessary to describe deep the methodology to determine cell membrane permeability.

Please, it is necessary to describe better MDA content determination. 

Line 147. Please adjust the chemical formula H2O

Please describe better the methodology used by LOX and PLD activity determination.

How did the authors validate the Primers used? 

Which housekeeping genes were used?  

Which was or were the statistical model used in work?

Results

Please, the authors need to deepen the analysis of Figure 1. Which is the difference between treatments? It is important not to include in the text that can be readily observable in the 

Figure 1. How was the severity of the chilling injuries in tested fruits?

Please, include the standard deviations in Figure 1. 

I suggest that the authors change the colors to the n butanol treatments or change the circle for another form where the difference is most noticeable for Figure 3-8a.

Discussion 

It is fundamental to explain o hypothesize why the fluctuations in the MDA, PA, and palmitic acid content values on days 14, 21, and 21, respectively, and on other days where the n-butanol treatment presented the worst value than the control treatment. 

The authors need to compare the results with more others' works and analyze their results. 

About the results obtained, Can the authors recommend an optimal time for melon storage treated with n-butanol?

Best Regards

Author Response

Thanks to the reviewers for their comments, we will reply to your comments in detail in the following article.

Round 2

Reviewer 1 Report

The authors made all the corrections. The paper in the present form is suitable for publication.